# Revisiting Leadership in Schools: Investigating the Adoption of the Dubai Inclusive Education Policy Framework

**Ayman Massouti** [1], **Nessrin Shaya** [2] **and Rawan Abukhait** [3,*]

1   Department of Education, College of Arts & Sciences, Abu Dhabi University, Dubai Campus,
    Dubai P.O. Box 410896, United Arab Emirates
2   Department of Education, College of Education, American University in the Emirates,
    Dubai P.O. Box 503000, United Arab Emirates
3   Department of Management, College of Business Administration, Ajman University,
    Ajman P.O. Box 346, United Arab Emirates
*   Correspondence: r.abukhait@ajman.ac.ae

**Abstract:** This study examined the prevalent leadership practices in the implementation of the Dubai Inclusion Education Policy Framework (DIEPF), which paves the way for elaborating on key nuances relating to the prospects and challenges of meeting the United Arab Emirates' (UAE) federal requirement of educating all students, including students identified as experiencing special educational needs and disabilities (SEND) in a common learning environment. A qualitative phenomenological research design was employed, where data was collected through semi-structured interviews with a group of individuals who assumed a senior leading role in SEND departments, led inclusive practices at the school level, or were experts in special education studies in the UAE. The final sample consisted of 21 interviewees, of which 11 are heads of SEND departments, 6 are senior executive leaders at the designation of principal or vice principal, and 4 are experts in special education employed in reputable higher education institutions in the UAE. The analysis resulted in identifying five themes as critical key factors in inclusive education implementation and several sub-theme components. As an additional dimension, an emergent theme was derived, expanding the theoretical boundaries of the DIEPF. Finally, theoretical and managerial implications for effective inclusive practices in school settings are offered.

**Keywords:** leadership; inclusive education; schools; policy implementation; Dubai

## 1. Introduction

Inclusive education has become a global approach for the education of all learners in schools, reflecting a commitment to the international policy principles of equity and diversity in education [1,2]. Among those policies are the United Nations Sustainable Development Goals (SDGs) and the Convention on the Rights of Persons with Disabilities that have called upon countries to engage in education reforms that would further acknowledge the needs of people with disabilities within schools and society [3]. In the United Arab Emirates (UAE), the response to these calls has been represented in the release of many inclusive education policy documents by the governing education institutions, including the Ministry of Education (MoE) and the Knowledge and Human Development Authority (KHDA) [4–6]. Further, most of the policy research conducted in this area seems to have focused on pre-service and in-service teachers' perspectives towards the inclusion of students with special educational needs [7–12]. Hence, the literature of inclusion policies in the UAE lacks studies that address the perspectives of school leaders in Dubai on the implementation of these policies and how students with special education needs, termed alternatively as Students of Determination (SoD) in the UAE, are supported [5]. Acknowledging their valuable contribution to the UAE's society and recognizing their strengths and capacities, persons with disabilities have been named "People of Determination" by

his Highness Sheikh Mohammed Bin Rashid Al Maktoum, the ruler of Dubai [5]. One of the challenges of implementing inclusion in the UAE's public schools pertains to the gap between policy guidelines and the attitudes and beliefs of policy actors, namely school leaders according to this current study [7]. In contrast, in the private sector, many schools in the UAE fall short in their budget for hiring more employees who would offer more tailored support and services for SoD [13]. While many studies about inclusive education have been conducted in the UAE within the country's public sector, there is a lack of studies conducted within the private sector and particularly in Dubai where most schools are private and governed by the Knowledge and Human Development Authority (KHDA).

Emphasizing the crucial role of school leadership and the overall environment established in schools, the KHDA 2019's guide stated that it is imperative for school leaders to "continuously monitor the expression of beliefs and attitudes and evaluate their impact upon lowering barriers to the achievement of inclusive education" [5] (p. 10). In Dubai, students are identified with special education needs when the impact of a disorder or an impairment becomes a barrier to learning, restricting the student's ability to enjoy equitable learning opportunities alongside their peers, hence an Individualized Education Plan (IEP) becomes a significant tool to ensure a quality educational provision for SoD [5]. Furthermore, in Dubai's private schools, inclusion is viewed as a collective practice where the school principal creates and leads an inclusion support team. This team consists of an inclusion champion, leaders of provision for SoD, representatives from both Support Teachers (STs) and Learning Support Teachers (LSTs), where each team member has a specific role to perform towards achieving the vision of inclusion [5].

### 1.1. A Critical Perspective towards Inclusion

Inclusive education is a process that pertains to identifying and removing all those barriers that hinder the participation and achievement of learners, exclusively those at risk of being excluded and marginalized [14]. Further, inclusion issues have become more complex as the inclusive education philosophy is changing from the focus on supporting students with disabilities to the education of all learners, and consequently, to general education [15]. This "focus must not simply be on access to general education, but rather the assurance that when inclusion is deemed appropriate, it is implemented with proper attitudes, accommodations, and adaptations" [15] (p. 287). However, inclusion remains "a complex and contested concept and its manifestations in practice are many and various" [16] (p. 3). Due to the ambiguity of the concept, it becomes important to address inclusion and its principles from an evidence-based approach by revisiting the leadership practices associated with the enactment of inclusion policies and how these practices are modified [16].

### 1.2. Research Context

Dubai is one of the seven emirates that constitute the country called the United Arab Emirates—an ethnically diverse, multicultural, and multi-religion country located in Western Asia [17,18]. More than 90% of Dubai's population are expatriates, hence the existence of 17 international curricula that are implemented in the emirate's large body of private schools [5]. Dubai operates 215 private schools that serve 289.019 students among which 30515 are Emiratis, all representing a total of 185 nationalities as of November 2021 [19]. In contrast, UAE government schools are comprised of mostly Emirati students (80%) with 20% of other nationalities, mainly those of Arabic descent [17]. In a report issued in 2019 by the National News in the UAE, 6% of Dubai schools' students have special education needs, particularly physical and intellectual disabilities. Some of the challenges faced by these learners are educational provisions, teacher professional development for inclusion, leadership in schools, and admission criteria, among other factors [5].

The UAE has enacted several laws that pertain to inclusive education. These are Federal Law 29 of 2006; Federal Law 116 of 2009; and School for All Policy of 2010; however, the KHDA in the Emirate of Dubai, as the governing authority of private education, has

released various inclusion-related policies that are tailored to its schools' context [4,5,20]. For this study, the aim was to investigate the adoption of the Dubai Inclusive Education Policy Framework (DIEPF) by Dubai schools' leaders and attending to the perspectives of inclusion experts in the UAE [4]. Exploring the perspectives and practices of these leaders while considering the above-mentioned policy will offer insights on how to move forward to establish a more robust inclusive culture and practices that support SoD in schools. Furthermore, this exploration will contribute to the enhancement of Dubai private schools' strategies in relation to teacher and staff professional development for inclusion, raising more awareness about inclusion support and services among school personnel and families of SoD, as well as informing future updates and implementation processes of school-based, inclusion-related policies. As such, this study aims at addressing: (1) how do Dubai's private school leaders conceptualize inclusion in their situated school context; and (2) how do they translate the Dubai Inclusive Education Policy Framework's (DIEPF) standards into their practices, and what challenges (if any) does this translation entail?

## 2. Literature Review

The review of the literature examines previous studies of inclusive leadership practices, shedding light on the various policies issued, their aims, and objectives along with key findings identified from UAE-based and international studies about inclusive education and related policy implementation.

### 2.1. Inclusion Policy Implementation: International Perspectives

The inclusive education movement has been and continues to be recognized as an international leading force towards the advancement of education policy and practice [21]. For instance, the enactment of inclusive education policies in South African schools is complex due to long-held beliefs that have fostered exclusionary practices for years [22]. Highlighting the role of context in policy implementation, inclusive education policy in South Africa did not develop in line with the pedagogical revolution but rather got "stuck at a political level since it ignored epistemological issues in the training of educationists" [22] (p. 2). A recent study in Serbia found that the inadequate teaching and learning conditions in schools along with poor professional competencies constitute major reasons for ineffective inclusive policy implementation [23]. Reducing class size, engaging teachers in more inclusion-oriented curriculum planning in schools, adapting teaching content, and seeking support from experts in inclusive pedagogies could be some of the strategies to advance inclusion [23]. In Korea, a competitive learning environment exists, as academic achievement is of high concern among parents [24]. This has created challenges for enacting inclusive education, as Korean students are under severe pressure due to their parents' high expectations [24]. To successfully enact such policies, school leaders need to exercise more flexibility and understanding of the inclusive dimensions and actively participate in supporting struggling learners [24].

It was argued that "despite the importance of schooling that meets all children's needs and maximizes their potential regardless of ability, inclusive education is still a contentious area of education policy" [25] (p. 1078). Relatedly, the ways in which Irish inclusion policies are being incorporated into practices remain contextualized and subject to the multiple interpretations of policy actors in schools [26]. Consequently, students with special education needs (SEN) in Ireland continue to move from mainstream schools to schools that only cater for students with SEN due to inadequate leadership and school environments that foster exclusion [26]. Hence, the enactment of inclusive education has to overcome many obstacles, including lack of teacher training and effective leadership, as well as an irrelevant curriculum structure and resources [26]. In Hong Kong, the lack of teachers' autonomy and inclusion-related experience along with fixed curricula and high working demands have collectively obstructed the implementation of effective inclusive education [27]. To mitigate these inclusion- and leadership-related challenges in Hong Kong, it is believed that the external control on students' achievement such as testing

requirements must be minimized, whereas teachers' autonomy in supporting diverse learners needs to be maximized [27].

## 2.2. Leadership in Schools: An Inclusive Approach

Inclusive leadership entails advocacy practices for inclusive education, raising awareness and teaching school community members about the rights of students with special education needs, establishing an ongoing dialogue about inclusion, as well as developing policies that support inclusive practices and students' learning outcomes [28]. Theoretically, an integrative inclusive leadership framework would consider "individual and collective effectiveness as well the importance of participation, engagement, safety, voice, and equity in the context of the complexity and intersectionality of multiple social identities, intergroup relations, and their multifaceted organizational and societal manifestations" [29] (p. 4). Consequently, fostering and sustaining an inclusive learning environment in workplaces such as schools become undoubtedly critical pillars of leadership in the 21st century [29].

There are few studies that have focused on investigating leadership that enhances inclusive practices in schools in Arabic-speaking countries and how school leaders can create an inclusive school culture [30]. In schools that promote inclusive culture, leadership is concerned with the adoption of collaborative approaches and interpersonal relationships among the individuals involved in critical work processes, such as those that pertain to inclusive education [31]. Therefore, it becomes crucial to have school leaders who possess inclusion-related knowledge and skills to support diverse learners and the effective implementation of inclusive education policies. Thus, inclusive leadership does not exist "in a vacuum, but rather in a space where families, students, teachers and administrators bring complex histories and experiences which principals must therefore navigate" [32] (p. 515). This resonates with other studies that called on school leaders to work with both special and general education teachers as well as other school professionals to support inclusion implementation and identify any organizational conditions that may be complicating inclusive teaching, programs, and services in schools [32].

Undoubtedly, school leaders in different contexts are responsible for developing a culture that values and supports students with SEN [33]. To practice inclusion, these leaders must seek to promote access for all learners to the general curriculum through effective teaching and learning processes [34]. In addition, school leaders must promote the value of shared responsibility among teachers and school community members, and in turn elevate schools' capacity for inclusive practices [35]. That said, school leaders become either enablers or disablers of inclusive education depending on their school context and their own capacities, beliefs, and policy practices.

A recent study in the UAE on the role of school principals in promoting inclusive schools found that several challenges continue to exist. These include lack of professional development for school principals on how inclusive education relates to teachers' increasing responsibilities and to school efficiency in supporting diverse learners [36]. Relatedly, a study in the UAE revealed that "school administrators (principals, vice-principals, and academic principals, inclusive) play a core role in school development by evaluating school performance, raising standards, identifying areas for continuous improvement, sharing knowledge and advocating continuous professional development" [37] (p. 223). Consequently, the school leader in the UAE becomes an agent of change who contributes to the implementation of successful educational programs such as those that pertain to inclusive education [38].

## 2.3. Inclusive Education in the UAE: Laws, Policies, and Practices

The UAE continues to invest in the field of inclusive education to support the inclusion of SoD into mainstream schools and ensure that their needs are well catered for by the various educational institutions. "One core objective for the implementation of inclusive education is to promote acceptance and peaceful co-existence between students with disabilities and typically developing peers in classrooms. Since 2006, the federal

government of the United Arab Emirates (UAE) has developed policies aimed at achieving inclusive education at all academic levels" [39] (p. 1). This aim is reflected in many policy documents and legislations such as Ministerial Resolution No. 647 for the year 2020 on the policy of inclusive education, which guides government schools in the UAE to adopt the necessary changes needed to ensure SoD have access to quality education and needs-based services [40]. Furthermore, the MoE has established the Department of People of Determination (formerly Department of Special Education) to protect the rights of SoD and ensure their access to a range of supporting services that facilitate their involvement in schools and society. These services include, but are not limited to, professional training for teachers of SoD in different areas such as Braille and sign language, hiring expert teachers specialized in behavior analysis and skill development, as well as developing support centers across the UAE to monitor the progress of SoD before and after their integration into mainstream schools. These centers seek to support parents of SoD in dealing with their children, providing recommendations on available special education services, and performing diagnostic assessments of disability for SoD [40].

Arguably, inclusive education in the UAE has been significantly prioritized within the government's policies and the various educational strategies developed [5]. Reflecting its commitment to inclusive education, the UAE in 2010 ratified and adopted the United Nations Convention on the Rights of Persons with Disabilities (UNCRPD). Years later, the UAE Vision 2021 embodied the country's national priorities and initiatives for the education field, laying the groundwork for the "UAE Centennial 2071". Among the priorities of the 2071 vision is education excellence as an empowering tool to develop the UAE's future generations' capacities for building a more cohesive and knowledgeable society that values and acts upon the principles of equity, diversity, respect, and acceptance [5].

*2.4. Inclusive Education Practices in Dubai*

A few years earlier, two Dubai-based pieces of legislation that pertain to the inclusion of SoD have been enacted [5]. These are the Dubai Law No. 2 of 2014 concerning the protection of the rights of persons with disabilities, and Executive Council Resolution No. 2 of 2017 that is concerned with the regulation processes of Dubai private schools. These pieces of legislation were sought to enforce the rights to equity education and full accessibility for SoD within Dubai's private education sector.

The DIEPF (2017) came to resonate the objectives of the United Arab Emirates' local laws and legislation, namely Federal Laws of 2006 and 2009 as well as Dubai Law No. 2 of 2014 that call for the inclusion of all learners. Moreover, the framework constitutes a commitment of the UAE to the United Nations Convention on the Rights of Persons with Disabilities (UNCRPD) [4]. According to the DIEPF, "it is the progressive development of attitudes, behaviours, systems and beliefs that enable inclusive education to become a norm that underpins school culture and is reflected in the everyday life of the school community" [4] (p. 53). Given the complexity of the Dubai educational landscape, this framework becomes a necessary document in assisting leaders in various school settings in Dubai mitigate the various barriers that act as a limiting factor for an equitable learning experience for all. Indeed, the framework is intended as a policy guide not only for schools but also for those working in various education facilities, including special education centers, early learning centers, and higher education institutions.

The DIEPF sets clear standards for the inclusive education provision to guide Dubai's education providers towards enhancing their inclusive education-related programs and services. These standards are: (1) Identification and Early Intervention; (2) Admissions, Participation, and Equity; (3) Leadership and Accountability; (4) Systems of Support for Inclusive Education; (5) Special Centers as a Resource for Inclusive Education; (6) Co-operation, Co-ordination, and Partnerships; (7) Fostering a Culture of Inclusive Education; (8) Monitoring, Evaluation, and Reporting; (9) Resourcing for Inclusive Education; and (10) Technical, Vocational Education and Training (TVET), Higher Education, and Post-School Employment [4] (p. 12). For this study, the standards 5 and 10, namely Special Centers as a

Resource for Inclusive Education and Technical, Vocational Education and Training (TVET), Higher Education, and Post-School Employment, respectively, have been omitted as they are beyond the scope of this study.

The DIEPF lays out these 10 standards along with the necessary actions needed to be taken by the various education institutions in Dubai to include SoD [4].

To ensure successful and promising inclusion, the DIEPF urges these institutions to establish monitoring processes and mechanisms that examine the effectiveness and the quality of implementation of these standards. Similarly, KHDA emphasizes the necessity to serve SoD in Common Learning Environments (CLEs). CLEs have been defined as "an educational setting where students from different backgrounds and with different abilities learn together in an inclusive environment. Common learning environments are used for the majority of the students' regular instruction hours and may include classrooms, libraries, gym, performance theatres, music rooms, cafeterias, playgrounds and the local community. A common learning environment is not a place where students of determination learn in isolation from their peers" [5] (p. 37).

## 3. Methods

We utilized a qualitative phenomenological research design to explore the prevalent practices in the implementation of the DIEPF. The principal method of data collection was by means of semi-structured interviews, with a group of individuals who assumed a senior leading role in SEND Departments, led inclusive practices at the school level, or were experts in special education studies in UAE. The guiding principle in the construction of the interview protocol was the DIEPF itself. Accordingly, three different sets of questions were developed for the three groups of participants.

Sample questions for head of SEND departments included:

1. Based on your knowledge, can you please tell me a little bit about how early identification of students of determination in your school takes place? What kind of assessments/strategies do you use for identification purposes?
2. Please take me through the Inclusion Support Team (IST) in your school. Who is on that team, and what do their various duties look like?
3. What qualifications do you and your team members have in relation to inclusion?
4. Looking at the admission process for Students of Determination (SoD) in your school, how do you fulfill this policy standard? Do you face any admission challenges? If yes, what are they?
5. What strategies do you follow to engage other school community members (parents, medical practitioners, social workers, etc.) towards supporting SoD?
6. What are the challenges facing your Inclusion Support Team (IST) while working with classroom teachers? How do you overcome these challenges?

Sample questions for principals and vice-principals included:

1. Please describe the cultural environment on your campus. How inclusive is it? Provide some examples.
2. What are the main qualities of a leader for successfully implementing the framework standards?
3. Have you received any diversity, inclusion and or cultural competence training? If yes, how have you applied what you learned on the job?
4. Can you please further elaborate on your leadership style.
5. According to research, teachers are facing a lack of support, training, and the needed resources for inclusion. How are you dealing with these issues in your school?

Research questions that strive to reach an in-depth understanding of patterns of individuals' practices and behaviors call for a qualitative research design [41]. Participant selection capitalized on utilizing the expert knowledge of executives affiliated with leading private schools in Dubai and universities offering Bachelor of Education programs. Accordingly, purposive sampling took place. Data were collected during the Spring 2021 and Fall

2022 semesters, through face-to-face and online semi-structured interviews, lasting around 30–50 min each. Subject recruitment was based on researchers' personal connections. Prior to enrollment, an invitation email to take part in the study was sent to potential subjects. Information pertinent to the aims and objectives of the current research study and a consent form encapsulating the subject's voluntary participation, confidentiality, anonymity, and the right to withdraw at any time was shared. Upon receiving confirmation of participation, subject names were given codes and listed in table form for future use in the analysis. Subjects were mainly contacted directly through their personal email accounts and, in a few cases, through the office of their executive assistants. Semi-structured interviews were employed, facilitating an inductive research approach and allowing the researchers to examine the issue in question at microlevels [42]. Semi-structured interviews are considered as an important tool for gathering data, as these interviews provide a relaxed and casual setting for subjects to express and reflect on their experiences, while also enabling interpretation of data, validity checks, and triangulation. These interviews provide a platform for subjects where they may easily express difficult issues and explicate or clarify questions [43].

In total, 23 highly experienced participants holding related roles of inclusion in their academic institutions were invited to participate, while two declined during the study due to unavailability. The final sample consisted of 21, of which 11 were heads of SEND Departments (52%), 6 were senior executive leaders at the designation of principal and vice principals (29%), and 4 were experts in special education (19%) employed in reputable higher education institutions in the UAE. As such, most participants were heads of inclusion departments (SEND). Of the 21 interviewees, 13 were females, 18 interviewees held postgraduate degrees, and the majority had between 11 and 15 years of experience. The age groups ranged from 37 to 55 years old. Interview participants included 14 females and 7 males, and the majority had around 11–15 years of experience (48%) (Table 1).

**Table 1.** Demographic characteristics of the respondents.

| Characteristic | | Value |
|---|---|---|
| Gender | Male | 7 (33%) |
| | Female | 14 (67%) |
| Educational Background | PhD | 5 (24%) |
| | Masters' Degree | 13 (62%) |
| | Bachelor's Degree | 3 (14%) |
| Age | 30–40 | 8 (38%) |
| | 41–50 | 12 (57%) |
| | 50+ | 1 (5%) |
| Designation | Experts in Special Education | 4 (19%) |
| | Head of Inclusion | 11 (52%) |
| | Principle and Vice Principles | 6 (29%) |
| Years of Experience | 0–10 | 8 (38%) |
| | 11–15 | 10 (48%) |
| | 16+ | 3 (14%) |
| Nationality | Non-Emirati | 21 (100%) |

Interview recordings were professionally transcribed, and data were analyzed via theoretical thematic analysis in which the data were organized around major findings [44]. Once the transcripts were ready, the themes were identified using a six-phase approach: familiarization with data, generation of initial codes, searching for themes, reviewing themes, defining and naming the themes, and producing the report [44]. Emerging codes were regarded as potential themes in a process where various codes merge toward forming overarching and, eventually, themes, sub-themes and subtheme components. Throughout all phases, constant cross-checking of data extracts, codes, and themes against each other and the entire dataset was performed.

The authors conducted all interviews and generated initial codes and themes. The theoretical thematic analysis is primarily analyst-driven, which was deemed as highly

relevant for this study in the light of the research questions. Accordingly, the thematic areas in this study were driven by the literature of inclusive education and the DIEPF, with the subtheme components subsequently emerging from the data.

## 4. Results

This section outlines and analyzes the prevalent practices in the implementation of the Dubai Inclusion Policy Framework, which paves the way for elaborating on key nuances relating to the prospects and challenges of meeting the federal requirement of educating all students, including students identified as experiencing special educational needs and disabilities (SEND) in a common learning environment. In such settings, all students have access to quality instruction, intervention, and support, so that they experience success in learning. Accordingly, five key themes presenting as key factors in inclusive education implementation emerged, namely: (1) Organization Dimension Factors in Implementation; (2) Administration and Operations' Dimension Factors in Implementation; (3) Practical Problems for Parents Associated with Implementation; (4) Practical Problems for Schools Associated with Implementation; (5) Expanding DIEPF to include Community Partnerships and Engagement. Several sub-theme components were also identified (See Table 2 for details).

**Table 2.** Themes and subthemes of data analysis.

| Theme | Subtheme |
|---|---|
| Organization Dimension Factors in Implementing Inclusive Education | • Leadership and accountability.<br>• Fostering a culture of inclusive education.<br>• Resourcing for inclusive Education. |
| Administration and Operation Dimension Factors in Implementing Inclusive Education | • Extensive systems of support for inclusive education.<br>• Admissions, participation, and equity.<br>• Monitoring, evaluating, and reporting impacts of individualized inclusion programs.<br>• Identification and early intervention.<br>• Parents' engagement. |
| Practical Problems for Parents Associated with Implementing Inclusive Education | • Discriminatory fees and expenses.<br>• Ratio of special need students to regular education students.<br>• Parental and community involvement: The school is not used as a community resource. |
| Practical Problems for Schools Associated with Implementing Inclusive Education | • Struggles to meet the costs of special needs support.<br>• Inventories for special education services and facilities as inclusive networks between schools.<br>• Preparedness of school leaders for inclusion.<br>• Availability of sufficient material resources. |
| Expanding DIEPF to include Community Partnerships and Engagement | • Federal funding to finance special needs education in mainstream schools.<br>• Strengthening partnerships with humanitarian organizations to fund additional costs.<br>• Strengthening partnerships with donor support groups to fund schools.<br>• School–community learning partnerships to spread awareness and support student success.<br>• Using collaborative school partnerships to foster and support inclusion. |

### 4.1. Theme 1: Organization Dimension Factors in Implementing Inclusive Education

All the interviewees indicated the importance of management and organizational dynamics in (1) the effective implementation of the DIEPF, (2) identifying leadership and

accountability, (3) fostering a culture of inclusive education, and (4) resourcing for inclusive education as main sub-themes, confirming the implementation of Standards 3, 7 and 9.

Considerable evidence pointed to the managerial roles and responsibilities depicted through leading, planning, organizing, and controlling. Leading was demonstrated through leadership skills manifested in employing inclusive leadership approaches and instructional and transformational leadership. Interviewees confirmed the eradication of some of the bureaucratic barriers that are seemingly put in place to slow down the rate of proper early identification despite the financial costs. The presence of adequate planning and strategies was evident, through development and dissemination of inclusion policies and strategic inclusive education improvement plans. Organizing people and materials was demonstrated through appropriate human resources that ensure that the school has sufficient and competent human resources to respond to diversity, and academic staff has access to expertise for further support. Finally, controlling is related to monitoring and evaluation. As one of the interviewees mentioned:

> *I don't know if you're familiar with it, the London Leadership Strategy team in the UK created what is called the SEND whole school-approach framework. And it's basically a quality audit framework for schools. Yeah, so we used it in the UK, and I've started to draft something similar for our schools so that they can actually, if we put schools in a cluster, we can say as heads of inclusion, go and visit each other and do like a quality assurance visit yourself. Yeah. And let's see, let's share that practice. Let's see how you can provide and support each other.*

There is enough evidence in the collected data to suggest that promoting a culture of diversity and inclusion among students and staff is a priority for the schools. This approach aims at the prevention of discrimination, nurturing culturally intelligent citizens and fostering effective interaction and communication with students of different ethnic and religious backgrounds. Particularly, school practices were characterized by: (1) schools enrolling students from different cultures, backgrounds, and nationalities, to promote diversity; (2) teachers and leadership promote values associated with the idea of inclusion; teachers see the prevention of discriminations as part of their teaching.

The respondents confirmed offering a variety of resources to improve and support diversity and inclusion efforts across their schools. Three main types of resources were identified, namely, training resources, technological resources, and material resources. Interviewees explicitly pointed out that many factors come into play in the design of truly inclusive classrooms and in supporting the inclusive and diversion notion of schools. In that sense, all interviewees concurred on providing extensive training and continuous professional development opportunities to support either teachers in general or particularly teachers of in the SEND department. As two of the interviewees explained:

> *"Regarding general teacher training, students deserve the right to be engaged, and no child should be left out, yet it becomes equally challenging as a teacher to leverage students' needs and instruction. Therefore, training focus on creating an inclusive classrooms, laws, and policies, building on a child's strength, what the curriculum content of an inclusive classroom is, and improving teaching skills for those with severe and complex needs and behaviour. The technological resources at the addressed schools by the respondents appeared to be appropriate for the diversity and number of the students";*
> *" . . . Similarly, the school's installations were accessible to all students, ensuring the presence of physical resources".*

*4.2. Theme 2: Administration and Operation Dimension Factors in the Implementation of Inclusive Education*

Enough evidence suggests that the overarching educational system within schools is designed to suit the needs of all learners, and particularly students exposed to SEND. The school administrators, personnel, and teachers serve as a pivot around which the educational programs revolve. Such educational systems are particularly characterized by (1) extensive systems of support for inclusive education; (2) equal opportunities in

education through admission and participation; (3) early identification and intervention of enrolled students; (4) monitoring, evaluating, and reporting impacts of individualized inclusion programs; and (5) parent engagement, reflecting implementation of Standards 1, 2, 4, 8, and 6.

Responses unraveled strategic approaches presenting as the underlying mechanisms of support for inclusive education, which aims at improving the quality of inclusion within schools. Such approaches capitalize on the broader idea of personalized learning, demonstrated through (1) the adoption of specialized multi-level delivery models; (2) the presence and availability of accommodation programs through adapted curricula and alternative plans; (3) development of IEPs; (4) teaching planning consideration of all the students that accommodate differences in abilities and learning styles. As a number of interviewees mentioned:

> *"We use Level System as reference to the level of provision that will be required for the students"; "Through our multi-level provision model, we extend our support for students of highest needs through adapted curriculum or school readiness support"; "We provide alternative paths where students in senior years leave with qualifications that offer flexible ways of developing and accrediting personal, social, and work-related abilities"; "IEPs comprise elements that document how student's progress toward meeting announced goals are measures, mainly through using objective/numerical scores, behavioural checklists, progress monitoring probes, and test scores".*

Interviewees attested that all SoD are entitled to the same entrance, participation, and equitable rights as all other students. They have the right to engage and take part in high-quality learning experiences with their peer group of students. Once enrolled, students experiencing special needs will receive active support to engage in the learning process as they reach their full potential and form bonds with their peers through social interactions in surroundings designed for common learning at their age. As some of the interviewees mentioned:

> *"Students of determination aren't denied admission based only on their history of need"; "The inclusive support team will make sure that students who experience special needs receive the assistance and needed accommodation"; "Curriculum alterations are necessary to provide fair access to educational opportunities".*

The findings showed clear evidence that schools' commitment towards ensuring the progress of students is closely tracked for impact. Accordingly, close monitoring, evaluating, and reporting took place, demonstrating regular and systematic collection and monitoring of student data, all of which allowed the IEP team to evaluate the appropriateness of the student's IEP. Relatedly, this theme is demonstrated through developing progress-monitoring plans and conducting self-evaluations. In particular, interviewees confirmed the importance of conducting self-evaluations for SEND and IEP teams, which help them engage in a reflective practice on their own actions. As a number of interviewees mentioned:

> *"We do self-evaluations and all our schools have got action plans"; "We monitor the implementation of inclusion programs and provide periodic reports on the status of these programs and services"; "The impact of our inclusion provision is monitored and evaluated"; "SEND teams employ a monitoring and reporting strategy that is effective and user-friendly, to inform decisions on the effectiveness of the allocated programs in helping students make progress"; "Students' progress against IEP goals are measured frequently and systematically, while these periodic reports on that progress will be provided to parents"; "Our self-evaluations aim at optimizing student outcomes for students with IEPs, determining compliance, and improving implementation of special education policies, procedures, and practices".*

All the interviewees revealed accurate identification and early intervention practices, which are well regarded as a critical component in the SEND provision. Identifying SEND in situations where it does not exist runs the risk of incorrectly labeling a student and driving

the unintended effect of reducing the compounded level of support given to those who might need it more. An early response to concerns, early identification, and intervention are key to helping children reach their potential. Accordingly, interviewees confirmed responsiveness to initial concerns about a child's progress and the identification of special educational needs, through (1) identification carried out through clearly assigned processes following initial concerns; (2) information from parents; (3) observations within the setting; and (4) intervention through targeted plans. As some of the interviewees explained:

> *"Identification is carried out through our learning support flowchart"; "At the event where concerns surface on students' progress, SEND Dept collects and collate information, in collaboration with parents/care givers on the child's learning and development, within and beyond the standard setting of learning; practitioner and professional observations, formal checks, any more detailed assessment, any specialist advice; progress in the prime and key areas, mainly, communication and language, physical development, social and emotional development"; "Discussions with parents can give practitioners insights into a child's personality, feelings, or interests outside the setting, that may affect a child's behavior, progress or development and need to be taken into account in planning support"; "For us observation is a powerful tool for gathering information about a young child. Following the observation, it is important to analyze and reflect on the information"; "If there are significant concerns or potential SEND students, practitioners and professional expertise would develop a targeted and action plan to support the child, involving other professionals such as, for example, the setting's SENCO or the Area SENCO. The progress check summary must describe the activities and strategies the provider intends to adopt to address any issues or concerns".*

The Individualized Education Plan (IEP) serves as a foundational blueprint and roadmap for special education services that will be offered. Interviewees confirmed that the development and creation of an IEP is a collaborative process that capitalizes on input from key stakeholders, among whom is the parent. The implementation of an IEP requires an ongoing and collaborative conversation between schools and parents. Accordingly, schools conduct meetings with parents to write the IEPs, then, additional meetings take place to report on students' progress relative to IEP targets, and finally, phone conferencing with parents are carried out to increase and maximize participation in ongoing webinars.

*4.3. Theme 3: Practical Problems for Parents Associated with Implementation*

The implementation of the policy in Dubai schools is strewn with logistical, social, and economic pitfalls, which means that students experiencing SEND may still face a compromised school experience—if they can enroll at all. Insufficient funding for parents with special needs students and inclusive schools was revealed as the biggest setback to the implementation of major programs, in addition to the prevalent student-teacher ratio for special education students and the school not being used as a community resource.

Considerable evidence points to discriminatory fees and expenses that further marginalize children with disabilities from poor social and economic backgrounds. As three of the interviewees contended:

> *"Almost every school in Dubai, including my own, imposes additional financial charges on the families of students with SEND. This is clearly a discriminatory policy and should be ended immediately. Too often, lack of financial resource is a deciding factor in whether a student with SEND attends the school of their choice. In my view this is morally questionable"; "The special-needs educational institutes are expensive and beyond the pocket of middle-class expatriate families"; "I had several parents over the years who agreed that therapy is needed but besides the already high school fees they cannot afford ongoing therapy. Many parents wait for school breaks to travel home to do therapy in their home country. The same goes for educational psychologists. Many parents told me that they cannot afford the full assessment. Shadow teacher has the same issues".*

Interviewees revealed that the current prevalent student–teacher ratio for special education students is too low, which is far less than the known rule of thumb of 70/30 split between students with and without disabilities. In that sense, class size and case load policies are unable to cater for the increased need for inclusive education by parents. Interviewees confirmed that the social effect of disability had more of a profound effect on the individual with a disability than what could be seen as a disability. At the current time, the contributing role of schools in building an inclusive community is not apparent. One of the underlying reasons for this is the absence of inclusive social networks where the school is not used as a community resource and of community involvement from a policy perspective.

*4.4. Theme 4: Practical Problems for Schools Associated with Implementation*

Considerable evidence points to several factors that may significantly challenge schools in their obligations to meet the federal requirements. These challenges included struggles to meet the costs of special needs support, an absence of inventories and facilities such as inclusive networks between schools, a lack of preparation for inclusion among school leaders, and the limited availability of material resources to support inclusion.

Typically defined as the number of students with Individual Education Plans (IEPs), the reality versus the current trend of increasing caseloads revealed that insufficient funding from the regular authorities to support schools presents the biggest setback to further supporting the implementation of major programs. As three of the interviewees explained:

*"The major challenge is reaching the capacity of the inclusion team. Without increasing the number of the support staff there will be no time to provide help for all. Inclusion is responsible not only for the support SoD but also EAL (English as a second language) and gifted and talented"; "It is highly costly on the school to be able to cater for all cases, such as autism, developmental delay, emotional/behavior disturbance, intellectual disability, orthopedic impairment, learning disability, learning disabilities and speech and language impairment"; "If you said to GEMS or all of our private school providers, you must spend, so we anticipate 10% of your school population will be SEND, you must spend 10% of your income on meeting the needs of children with SEN, right".*

Interviewees highlighted the absence of networks between schools that comprise specialist teachers that can then provide additional support to other teachers in terms of teaching strategies. As two of the interviewees explained:

*" . . . . and if we invest in our specialist teacher, we will then create a network of specialist teachers within our schools that can then provide additional support to our teachers in terms of teaching strategies"; "So, you know, it is a challenge for us, space is a massive, massive challenge for us in terms of having dedicated space for interventions and things like that".*

Principals' awareness of inclusive practices and education emerged as a significant factor in creating and promoting the right culture for inclusive schools. Absence of training in diversity, inclusion, and or cultural competence training, among school leaders specifically, was obvious within the interviews. On the other hand, the evidence is contradictory with respect to physical resources, where not all schools provide sufficient physical resources to respond to the needs of all their students. As two of the interviewees delineated:

*"Majority of our schools have got a sensory room"; "The sensory rooms, the chill out rooms, the therapy spaces, sensory playgrounds are available, but not in all our schools".*

*4.5. Theme 5: Expanding DIEPF to Include Community Partnerships and Engagement*

This paper draws attention to other important relevant factors in realizing inclusive education, namely, the way special needs education in mainstream schools is funded and the partnerships that promote inclusive education for students with disabilities. Accordingly, a new theme emerged that aims at expanding the DIEFP framework to include community partnerships and engagement with a number of sub-theme components: (1) Federal

Funding to Finance Special Needs Education in Mainstream Schools; (2) Strengthening Partnerships with Humanitarian Organizations to Fund Additional Costs; (3) Strengthening Partnerships with Donor Support Groups to Fund Schools; (4) School-Community Learning Partnerships to Spread Awareness and Support Student Success; and (5) Using Collaborative School Partnerships to Foster and Support Inclusion.

Special education should be supported by local and federal funding, to cover the additional tuition fees and relevant costs incurred by the needed special education services. Every child deserves the opportunity to succeed, and no child should be left behind. Yet, as mentioned earlier in the analysis section, enrolling special needs students within small groups, supported by highly trained teachers along with providing professional services, can often be expensive. Such discriminatory fees marginalize parents of special needs students. Accordingly, resources need to be mobilized to avail accessible financial grant services to all families of pupils with certified special educational needs and nationalities in the UAE that meet the criteria. Such a step would support families in the Emirates that require financial aid to subsidize the cost of the evaluations, therapies, and support needed to help students realize their full potential. Schools should partner with humanitarian organizations, foundations, and associations that serve children with special needs to support special needs education in mainstream schools through either providing or covering the costs of therapy and playground equipment, therapy toys, and other similar resources. Corporate giving increased by 8% between 2016 and 2017. Businesses are becoming more aware of the positive impact they can have on societal issues.

When schools and community organizations work together to support learning, everyone benefits. Partnerships can serve to strengthen, support, and even transform individual partners, resulting in improved program quality, more efficient use of resources, and better alignment of goals and curricula [45]. In that sense, schools could work in partnership with communities and organizations to support children's learning during afterschool hours and during the summertime. Consequently, there needs to be tremendous growth across the nation in intentional efforts to forge meaningful partnerships between schools and afterschool and summer programs, with particular focus on SoD. The benefits of school–community partnerships lie in raising awareness of diversity among learners and members of society, while supporting inclusion in educational settings at the same time.

Forming collaborative partnerships with external education agencies such as schools as a form of strategic alliance is critical to promoting an inclusive culture in the school and broader community and successfully reaching inclusive education outcomes. Accordingly, such practices can take the following form: (1) share human resources such as teacher specialists who can provide additional support to teachers in terms of teaching strategies and specialized pedagogies; (2) share inventories of special education services and facilities, such as sensory rooms and pools, which cannot be part of mainstream set-ups, as inclusive networks between schools; (3) exchange material resources, such as specialized adaptive equipment, assistive technologies, and rehab products.

## 5. Discussion

This study aimed to examine the implementation of the Dubai Inclusive Education Policy Framework by leaders of Dubai's private schools. We were interested in the affordances, challenges, and possibilities experienced by these leaders, to inform future policy development that pertains to educating Students of Determination (SoD) and to determine how to empower the access of the latter to quality and equitable learning experiences. Looking at the analyzed data, we identified the appropriate leadership style for implementing inclusion policies, namely inclusive leadership. This finding conformed to earlier studies that positioned inclusive leadership as a core element in developing schools that reflect inclusivity and commitment to all students' wellbeing, including those with SoD [28,46,47].

### 5.1. Inclusive Climate and Early Identification in Schools

Beyond the positive implications of adopting instructional and transformational leadership, it was revealed in the findings that inclusive culture requires leaders to ensure the existence of preventive measures for discrimination in all its forms among students, teachers, and the community at large. Inclusive culture is viewed as an "inclusive climate" that serves a key role in establishing common beliefs of inclusivity among those involved [33]. However, the findings suggest that the inclusionary practices of school leaders are subject to various factors such as training and attitudes towards inclusion, among others, which, if implemented, would lead to greater opportunities of robust inclusion in schools [34]. These findings conform to previous studies that viewed general teacher education certification programs as a driving force in supporting inclusion in schools [34,48]. As exemplified in the participants' responses and supported in the literature, the adoption of the inclusion policy framework in Dubai's private schools seems to prioritize individualized learning [49]. This approach to learning embodies students' individual needs; however, we argue that it requires school leaders who are equipped with the necessary tools for success in their school's inclusion journey. These tools include the availability of assessment tools for evaluating, monitoring, and reporting on issues of SoD, as well as alternative plans for the current school curricula.

An early response and identification followed by a quick intervention are keys to helping children reach their full potential [50]. Accordingly, interviewees confirmed responsiveness to initial concerns about a child's progress and the identification of special educational needs through (1) clearly assigned processes following initial concerns; (2) information from parents; (3) observations within the setting; and (4) intervention through targeted plans. Consequently, the development and creation of the individualized education plan (IEP) that constitutes one of the outcomes of the intervention appears to be a collaborative process that capitalizes on input from key stakeholders, among which are the parents. Hence, the implementation of the IEP requires an ongoing and collaborative conversation between schools and parents.

### 5.2. Towards Enhanced Funding and Networking Strategies for Inclusion

Insufficient funding for parents of SoD was revealed as the biggest setback to the implementation of major inclusion programs, conforming to previous studies that addressed the challenges of inclusion implementation in schools [14,51,52]. Furthermore, the findings showed that schools are not viewed as community resources due to the business nature of private schooling in Dubai. Thus, the existing discriminatory fees and expenses associated with special education as addressed by the study participants in these private schools continue to further marginalize children with disabilities who come from disadvantaged social and economic backgrounds. These findings suggest that building an inclusive school community may be elusive and far from reality. In other words, inclusivity in action may not be operative until all parents of SoD are able to secure a decent placement for their children in Dubai's private schools without feeling anxious about how to cover the additional school fees for special education services. For that valid reason, our study calls on developing more robust inclusive social networks that engage government, local, and charitable organizations that speak on behalf of the People of Determination. The hope is that these networks can bring about change in the processes of funding for special education services in Dubai's private schools where most students including those of Determination are non-Emiratis and come from low to middle socio-economic class families.

From our interviewees, it was evident that not all schools are well equipped to meet the federal requirements for inclusive education provisions as exemplified in the Dubai Inclusive Education Policy Framework (DIEPF). Some of the study participants highlighted that not all schools have the same physical resources including, but not limited to, sensory rooms, materials for special education classroom instruction, and assistive technology devices, as well as the human resources required. Conforming to previous international studies on the implementation of inclusive education and leadership challenges [25–27,36],

human resources were found associated with a lack of well-trained school leaders for inclusion and a smaller number of special education specialists in schools. These findings suggest the development of inventories and facilities for inclusive education and the creation of networks between schools where teachers and school leaders can further support each other's practices in relation to inclusive education teaching and administration [35]. Such networks would allow for cooperation and experience exchange on issues that pertain to instruction, inclusive competency among education professionals, and more importantly the creation of promising inclusive schools that work [28,29].

*5.3. Expanding the DIEPF: Community Partnerships and Engagement*

The analysis of our study findings highlighted a new emerging theme that would expand the dimensions or the standards of DIEPF, namely Community Partnerships and Engagement. Further, this theme may include, based on our study findings, several sub-theme components that were found significantly important for promoting inclusive education in schools, namely (1) federal funding to finance special needs education in mainstream schools; (2) strengthening partnerships with humanitarian organizations to fund additional costs; (3) strengthening partnerships with donor support groups to fund schools; (4) school–community learning partnerships to spread awareness and support student success; and (5) using collaborative school partnerships to foster and support inclusion. Below, we elaborate on how these sub-themes can be manifested to further enhance the implementation of inclusive education policy provisions in schools in ways that support families seeking a quality and equitable education for their children with special education needs or, as termed alternatively, Students of Determination (SoD).

Humanitarian organizations, foundations, and associations can support children with special needs through either providing or covering the costs of therapy and playground equipment, therapy toys, and other similar resources for inclusion in schools. Accordingly, corporate philanthropy, referring to the activities which companies voluntarily initiate that aim to manage their impact on society at large, is showing tireless generosity amid challenging times following the COVID-19 pandemic. As such, schools should harness the benefits of corporations' commitment to social responsibility and philanthropy, where such donations could be extended and utilized to provide both human and material resources required for students identified with special education needs. Consequently, expanding the conceptual dimensions of partnership in the Dubai Inclusive Education Policy Framework would serve as a key to strengthening, supporting, and even transforming individual partners, resulting in the improved quality of special education programming in schools and ultimately more efficient, aligned, and targeted use of the available resources [45].

## 6. Conclusions

The Dubai Inclusive Education Policy Framework (DIEPF) has been established to support the UAE's adherence to both local and federal legislation and to the United Nations Convention on the Rights of Persons with Disabilities (UNCRPD), mandating the inclusion of all students irrespective of their abilities in the education sector. We anticipate that the following recommendations would further empower regulatory authorities and governing bodies in Dubai to closely monitor progress and compliance: (1) develop clear policies on how many students with IEPs can be in a regular classroom, along with student–teacher ratios for students with special needs; (2) delineate clear assessment strategies for equality and diversity to ensure that SoD are given access to a broad, rich, and deep curriculum; (3) delineate clear assessment strategies for inclusion to ensure SoD have equal access to high-quality education; (4) induce the DIEFP framework to have stronger implications on the way the Dubai Schools Inspection Bureau (DSIB) conducts its inspections, so that the framework will benefit SoD and education providers that have high proportions of SoD will be able to be judged good or better.

Additionally, this research offers school leaders and regulatory bodies several worthwhile insights into the extent to which education providers in Dubai's private schools

are responding to the needs of SoD and the UAE vision of creating a fully cohesive and inclusive society. Further to current study findings, a few recommendations are in place to support optimizing the success of DIEPF implementation at the managerial level of the school. Firstly, there is a need for all school leaders, mainly school principals and vice principals, to possess inclusive education certifications and training. Second, in order to overcome some of the challenges associated with school admission of SoD, there is a need to establish inclusion campaign programs that are geared towards parents of SoD, so they become more open about their child's needs and build a sense of trust with schools. Such programs will reduce the challenges of parents not disclosing the status of their child, a fact that has been found to complicate the school admission process of SoD. Third, developing an online inventory of special education services and facilities for all Dubai private schools would assist the KHDA in SoD's school admission. Such a tool will offer the KHDA the ability to identify what services are offered by these schools and develop a strategic plan to accommodate all SoD cases accordingly. Fourth, there is an absence of an established network of certified and KHDA-approved professionals who can further assess students with special education needs. Following the completion of a full screening assessment test for students, those expected to have special needs will be reported to KHDA, which in turn communicates with their parents and refers them to the abovementioned network.

Undoubtedly, one of the study's strengths is the inclusion of participants from high-achieving institutions with high levels of financial and human resources. The organizational dynamics of these schools were highly structured and advanced, associated with curriculum, instruction, and assessment aligned with local and international standards, leading to high expectations for all students and from all staff. Furthermore, the use of qualitative research methods allowed for a thorough examination of the phenomenon. In contrast, the interviews were not restricted to a rigid and predefined structure, a fact that facilitated the expansion of the theoretical boundaries of DIEPF. Yet, the study's focus on high-achieving schools was a limitation. Therefore, the assigned sample utilized for this study may not accurately reflect the intended population. Consequently, the sample may be expanded to cover more varied categories of schools in future studies, with the possibility of developing a quantitative evaluation instrument. Additionally, aspects such as teachers' attitudes towards inclusion practices in schools and a comparison of how inclusion policies work in other countries might be explored.

**Author Contributions:** All authors have contributed to this work. Conceptualization, A.M.; methodology, N.S.; formal analysis, N.S., R.A., A.M.; investigation, A.M., N.S., R.A.; writing—original draft preparation, A.M., N.S.; writing—review and editing, A.M., N.S., R.A. All authors have read and agreed to the published version of the manuscript.

**Funding:** This research was funded by Ajman University, Ajman, UAE.

**Institutional Review Board Statement:** Ethical review and approval were waived for this study since the Knowledge and Human Development Authority (KHDA) that serves as the governing body for private education in the Emirate of Dubai does not require researchers intending to do education research to obtain an ethics approval. According to the KHDA website, researchers can directly reach out to potential adult participants in Dubai's private education sector for recruitment. Hence, to agree or disagree to participate in research is at the discretion of the potential participants after they have received an informed consent form and a letter of information about the study.

**Informed Consent Statement:** Informed consent was obtained from all subjects involved in the study.

**Data Availability Statement:** Not applicable.

**Acknowledgments:** We acknowledge the support provided by Ajman University to publish this work.

**Conflicts of Interest:** The authors declare no conflict of interest.

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
