# Peer review of "Revisiting Leadership in Schools: Investigating the Adoption of the Dubai Inclusive Education Policy Framework"

_sustainability, doi:10.3390/su15054274_

Round 1

Reviewer 1 Report

- the literature review mostly refers to the inclusive educational context of UAE and Dubai in particular. In my opinion, you should also review other studies that focus on the same theme as your research and at the same time, give a theoretical background of the concepts that you are using (educational leadership, inclusive education, etc)

- at the same time, you should look into more recent studies (2021 and 2022)

- the results are somehow hard to read. in my opinion you should present them in a quantitative form, maybe using a table and coding the most common themes that you have found in your interviews

- the discussion section is very well organised, clear and easy to read and understand.

Reviewer 2 Report

Thank you so much for the opportunity to review this interesting work, dealing with a hardly researched topic in an underexplored context.

Accordingly, the article could make an important contribution to decolonizing research in the field, see, regarding the actual discussion:

Bainazarov, T., Gilzene, A. A., Kim, T., López, G. R., Louis, L., Oh, S. Y., & Taylor, E. K. (2022). Toward Decolonizing Our Scholarship and Discourses: Lessons From the Special Issue on Decoloniality for EAQ. Educational Administration Quarterly, 58(5), 810-829.

Overall, I am in favour of this study but the main concerns I have are related to the absence of the discussion regarding leadership in part two (Literature Review).

Here, the thematization of leadership in general, in Dubai, and in relation to inclusion is completely missing. However, there is an international discussion on this topic, see, for example:

Ryan, J. (2006). Inclusive leadership and social justice for schools. Leadership and Policy in schools, 5(1), 3-17.

Wang, F. (2018). Social justice leadership—Theory and practice: A case of Ontario. Educational Administration Quarterly, 54(3), 470-498.

Also missing are discussions on culture-specific leadership aspects in Dubai and the emic-etic-tension in leadership research, see, for example:

Khalifa, M. A., Gooden, M. A., & Davis, J. E. (2016). Culturally responsive school leadership: A synthesis of the literature. Review of educational research, 86(4), 1272-1311.

Den Hartog, D. N., House, R. J., Hanges, P. J., Ruiz-Quintanilla, S. A., Dorfman, P. W., Abdalla, I. A., ... & Zhou, J. (1999). Culture specific and cross-culturally generalizable implicit leadership theories: Are attributes of charismatic/transformational leadership universally endorsed?. The leadership quarterly, 10(2), 219-256.

These listed referecnces are mentioned in part, but only in the discussion.

It is imperative that authors set a theoretical framework here at the beginning and refer to it at the end.

All other points in the manuscript are influenced by this and depend on this elaboration.

More detailed comments are:

1. How is leadership connected with implementing inclusive education in Dubai (more precisely the Dubai Inclusion Education Policy Framework (DIEPF)?

2. The paper is an interesting and relevant topic in the field, as we hardly know something about that from the Arab world.

3. It adds a new perspective to previous published studies.

4. The references are appropriate. But authors need to elaborate in leadership in general and especially with regard to the Dubai or Arab context in detail in their review.

5. Tables and figures are okay.

Reviewer 3 Report

Dear authors,
Your paper is timely, well-written and understandable and gives a lot of information about inclusive education in Dubai.

To make your work stronger, I would have expected a clearer context of the semi-structured interviews, perhaps stating more specific questions used during the interviews.

Also, it would be very useful in the discussion, although the research is in Dubai, to correlate your findings with research findings from other countries, as the struggle to implement inclusive education is international.

Round 2

Reviewer 2 Report

Thank you very much for revising the manuscript. All my comments have been addressed. The article is much better understandable and more coherent now. I have no objection to publication.